# Control-oriented Energy-Based Actionable World Model for Decision-Making and Process Control

## Abstract

We introduce the *Energy-Based Actionable World Model* (EBAWM), a hybrid world-modeling framework for industrial process forecasting and control that combines deterministic state-space dynamics with an energy-based transition critic. EBAWM is designed for long-horizon, high-stakes decision-making, where reliable recursive prediction requires both stable state evolution and principled uncertainty awareness. In contrast to modern deep time-series models—such as CNNs, RNNs, and Transformers—that operate primarily as input–output predictors, EBAWM maintains an explicit, recursively propagated state tied to physically meaningful system variables. This structure enables state correction, long-horizon simulation, and direct integration with Receding Horizon Control, model predictive control, and model-based reinforcement learning. The deterministic transition model provides a strong inductive bias for system identification by favoring explicit, Markovian, action-conditioned state transitions, thereby mitigating representation collapse, a common failure mode in energy-based learning. Uncertainty is captured through an energy function that evaluates the plausibility of action-conditioned state transitions, rather than by injecting stochasticity into the dynamics or relying on model ensembles. High-energy regions naturally indicate dynamically inconsistent or out-of-distribution behavior, yielding an interpretable uncertainty-aware signal without assuming a parametric noise model. Our contributions are: (i) we show that the geometry of the learned energy landscape encodes dynamical structure and stability-related properties, enabling uncertainty-aware forecasting and implicit control; (ii) we introduce a control-oriented world model that combines recursive, action-conditioned physical state propagation with energy-based transition evaluation, supporting online optimization and closed-loop decision-making; and (iii) we propose a simple and stable energy-based modeling design that avoids representation collapse by operating on a latent space shaped by a deterministic forecaster.

## 1 Introduction

Accurate forecasting of industrial and physical time-series data is essential for monitoring, control, and anomaly detection. Classical deep learning architectures—such as CNNs, LSTMs, and Transformers—offer strong function-approximation capabilities but typically operate in an input–output setting, predicting future observations from fixed historical windows. These approaches identify nonlinear input–output mappings without maintaining an explicit Markovian state. This stands in contrast to state–space or input–state–output formulations—where the state is measured or recursively propagated—which underpin modern control design, including nonlinear control, model predictive control (MPC), and filtering techniques such as the EKF Masti & Bemporad (2021).

The absence of an explicit state obscures the system's latent dynamical structure, yields internal representations with limited physical meaning, and restricts the model's ability to assess the plausibility or confidence of its predictions. In contrast, models that maintain an explicit latent or observable state evolving through a learned transition function provide a structured mechanism for long-horizon simulation, improved interpretability, and compatibility with control frameworks such as MPC and reinforcement learning (RL).

Because the state summarizes the system's evolution, it can be corrected online whenever new measurements arrive, stabilizing closed-loop operation and mitigating the error accumulation that affects purely input–output models. This capability naturally supports Receding Horizon Control (RHC), where the controller repeatedly updates the initial condition using the latest measurement or state estimate, computes an optimal control sequence, and applies only the first action before replanning. In this sense, state-based world models support both classical RHC and its interpretation as a moving-window forecasting scheme driven by updated state estimates.

Industrial applications impose stringent requirements on learned dynamical models, particularly regarding stability and uncertainty awareness. In real-world processes—such as cement production, waste incineration, or chemical processing—systems are typically operated in inherently stable regimes. Learned dynamics should preserve this stability; otherwise, latent or predicted trajectories can become physically implausible and unusable for long-horizon prediction or control. At the same time, overconfident predictions can drive systems into unsafe operating regions. As industrial datasets grow to months or years of multivariate sensor measurements, quantifying epistemic uncertainty becomes increasingly important as a signal for extrapolation beyond the data regime. Stability and uncertainty awareness are therefore deeply intertwined requirements for trustworthy industrial decision-making.

Classical and data-driven control theory emphasize robustness and formal guarantees in both open- and closed-loop settings Berberich & Allgöwer (2025). In contrast, many learning-based control approaches do not explicitly enforce global dynamical properties, instead focusing on identifying and avoiding regions where the model is unreliable Janner et al. (2019). From a control-theoretic perspective, this constitutes a weaker notion of guarantee, as it does not establish global stability or performance bounds.

In energy-based formulations, model unreliability is naturally encoded by high-energy regions, which act as implicit barriers in the state–action space. Model-based reinforcement learning (MBRL), by comparison, typically does not require the learned dynamics to be globally stable, unbiased, or physically consistent. The model mainly serves as a short-horizon simulator, and inaccuracies are tolerated as long as they do not degrade policy performance within the finite rollout horizon. As a result, stability in MBRL is often handled implicitly—through short rollouts, frequent replanning, or uncertainty-aware objectives—rather than by imposing explicit constraints on the learned dynamics.

Recent work highlights the importance of uncertainty awareness in model-based decision-making. The Robotic World Model (RWM) Li et al. (2025) employs an autoregressive dynamics model equipped with ensemble-based epistemic uncertainty estimation. Deep ensembles Lakshminarayanan et al. (2017) provide a scalable and widely adopted approach for uncertainty quantification, and the resulting uncertainty signal is incorporated directly into the reward function to discourage policies from entering unreliable regions. Similarly, Model-Based Policy Optimization (MBPO) Janner et al. (2019) mitigates compounding model error by restricting synthetic rollouts to short horizons and filtering them using an uncertainty estimator. Together, these approaches demonstrate that the usefulness of learned dynamics models depends not only on predictive accuracy, but also on the ability to recognize when predictions are unreliable.

**World Models**  World models aim to capture the dynamics of an environment to support simulative reasoning and decision-making (Dawid & LeCun, 2024; Ha & Schmidhuber, 2018). For control applications, a world model must support action-conditioned prediction and counterfactual evaluation, capabilities required for MPC and model-based reinforcement learning.

Joint Embedding Predictive Architectures (JEPA) advocate a non-autoregressive, non-generative encoder–encoder framework that predicts future latent states without reconstructing raw observations. However, as noted in recent critiques Xing et al. (2025), many practical instantiations—including the latent world model of Dawid & LeCun (2024)—remain autoregressive in operation, relying on recursive state propagation during deployment.

A central claim in the JEPA framework is that deterministic latent-state prediction is sufficient, and that decoding observations is unnecessary. This philosophy is exemplified by Time-Series JEPA Girgis et al. (2024), where learning is performed entirely in latent space. In contrast, for industrial process control, explicit reconstruction of future observations is essential. Predicted sensor trajectories carry direct physical

meaning and are required for monitoring, validation, constraint handling, and control synthesis. While latent representations are valuable for learning compact dynamics, they do not replace physically interpretable predictions in safety-critical industrial settings.

**Energy-Based Models** Energy-Based Models (EBMs) are a flexible class of generative models (LeCun et al., 2006; Ngiam et al., 2011; Carbone, 2024) that define probability distributions via an unnormalized energy function, assigning lower energy to more plausible configurations. A key challenge in training EBMs is the intractability of the partition function. Practical training strategies—including contrastive learning, score matching, and MCMC-based methods—are reviewed in Song & Kingma (2021). EBMs have been applied to probabilistic regression (Gustafsson et al., 2020) and trajectory-level planning (Du et al., 2019), where Langevin dynamics refines multi-step predictions. These approaches demonstrate that explicitly representing plausibility—rather than committing to a fixed parametric uncertainty model—can significantly improve robustness, particularly under distribution shift.

**Contributions and Overview** In this work, we propose the *Energy-Based Actionable World Model* (EBAWM), a hybrid world-modeling framework that explicitly separates *predictive structure* from *uncertainty evaluation*. Our first contribution is to show that the geometry of a learned energy landscape encodes meaningful dynamical structure and stability-related properties. By analyzing gradients and curvature of the energy function, we obtain an uncertainty-aware signal that supports forecasting, stability analysis, and implicit control via energy minimization.

Our second contribution is a control-oriented world-model architecture that combines recursive, action-conditioned physical state propagation with an energy-based transition critic

$$E_\theta(s_t, a_t, s_{t+1}),$$

which evaluates the plausibility of predicted transitions. Unlike approaches that use energy functions as implicit dynamics models Girgis et al. (2024), EBAWM retains an explicit state–space formulation, enabling stable long-horizon rollouts and direct integration with receding-horizon control (RHC), model predictive control (MPC), and model-based reinforcement learning.

Our third contribution is a simple and stable energy-based modeling design that avoids representation collapse by coupling the transition critic to a supervised deterministic forecaster. The deterministic prediction loss shapes the latent state space, inducing structured, Markovian, action-conditioned representations that provide a strong inductive bias for learning. This contrasts with purely self-supervised collapse-avoidance strategies such as LeJEPA Balestriero & LeCun (2025), while remaining fully compatible with energy-based learning.

Overall, uncertainty in EBAWM is represented through the geometry of the learned energy landscape rather than by injecting stochasticity into the dynamics or relying on model ensembles. This design yields an interpretable and uncertainty-aware world model suitable for online optimization and closed-loop decision-making in control settings.

## 2 Related Work

This review is not intended to be exhaustive. We primarily focus on the reinforcement learning literature, where model-based approaches are often developed and evaluated in highly experimental settings. In contrast, the system identification and control communities typically emphasize formal guarantees, interpretability, and stability, which motivates a different set of modeling assumptions. Our work aims to bridge these perspectives by drawing from recent advances in model-based RL while addressing requirements central to control-oriented applications.

In the context of model-based reinforcement learning, a reference is the probabilistic dynamics model introduced in Janner et al. (2019). Their approach, Model-Based Policy Optimization (MBPO), is built upon an ensemble of neural network dynamics models $\{p_1, \ldots, p_B\}$, where each member parametrizes a Gaussian

distribution with diagonal covariance,

$$p_i(s_{t+1}, r \mid s_t, a_t) = \mathcal{N}\big(\mu_i(s_t, a_t), \Sigma_i(s_t, a_t)\big).$$

Each model in the ensemble captures *aleatoric uncertainty* through its output variance, while *epistemic uncertainty* is handled via bootstrapping (Bickel & Freedman, 1981): different models are trained on different resampled subsets of the data. This decomposition of uncertainty is crucial for reliable long-term predictions, especially in regions where data are scarce and the learned model might otherwise be exploited by policy optimization.

Energy-Based Models (EBMs) provide a conceptually different approach to learning Markovian state transitions. Instead of predicting a parametric distribution, an EBM assigns an *energy* to each transition, where lower energy indicates higher plausibility. In this setting, *epistemic uncertainty* emerges implicitly: transitions in data-sparse regions naturally receive higher energy due to the lack of supporting evidence, while transitions in well-sampled regions receive lower energy. Thus, the energy landscape encodes both plausibility and model confidence without requiring an ensemble.

A particularly relevant formulation is the trajectory-level EBM introduced by Du et al. (2019), defined within a Markov decision process $\langle S, A, T, R \rangle$. Instead of modeling the transition function $T$ explicitly, they parameterize an energy function $E_\theta(s_t, s_{t+1})$ and interpret it as an unnormalized score for state transitions. A trajectory is assigned energy by summing the local transition energies, with lower values corresponding to trajectories more consistent with the data. Training proceeds via a contrastive objective that encourages real transitions to have lower energy than perturbed negative samples. In their method, the energy function itself serves as an implicit dynamics model: entire trajectories are optimized through iterative Langevin updates that descend the energy landscape, enabling model-based planning without an explicit transition predictor.

A related line of work is *Offline Transition Modeling via Contrastive Energy Learning* Chen et al. (2024), which proposes learning energy-based models of system transitions for offline reinforcement learning. In their framework, an energy function is trained contrastively to assign low energy to observed state transitions and higher energy to perturbed or out-of-distribution transitions. The resulting energy landscape is used as an implicit dynamics model, enabling uncertainty-aware planning without explicitly parameterizing a probabilistic transition distribution. However, despite motivating energy-based modeling as an alternative to probabilistic dynamics and ensemble methods, Chen et al. (2024) ultimately relies on an ensemble of multiple energy-based transition models during policy optimization. This reintroduces ensemble-based heuristics as a mechanism for uncertainty handling, rather than deriving uncertainty directly from a single learned energy landscape.

## 3 Energy-Based Models

A standard Energy-Based Model (EBM) (LeCun et al., 2006; Murphy, 2013) defines an energy function $E_\theta(x) \in \mathbb{R}$, where $x \in \mathcal{X}$ and $E_\theta$ is parameterized by a deep neural network. EBMs specify unnormalized densities in which lower energy corresponds to higher plausibility. The energy function induces an unnormalized density via the Boltzmann distribution:

$$p_\theta(x) = \frac{\exp[-E_\theta(x)]}{Z(\theta)}, \qquad Z(\theta) = \int \exp[-E_\theta(x)] \, dx, \tag{1}$$

where $Z(\theta)$ is the generally intractable partition function.

**Inference with EBMs** For dynamical systems, the basic unit of prediction is a state–action transition. We therefore define a transition-level energy function

$$E_\theta(s_t, a_t, s_{t+1}) \in \mathbb{R}, \tag{2}$$

which evaluates the plausibility of moving from state $s_t$ to $s_{t+1}$ under action $a_t$. Transitions that align with the true system dynamics should correspond to low energy, whereas implausible or out-of-distribution transitions receive high energy.

Under the Markov assumption, the next state depends only on the current state and action. This induces an unnormalized conditional transition density:

$$p_\theta(s_{t+1} \mid s_t, a_t) \propto \exp[-E_\theta(s_t, a_t, s_{t+1})]. \tag{3}$$

Extending this idea to full trajectories, Du et al. (2019) factorize the trajectory density over consecutive transitions. For a trajectory $\tau = (s_1, \ldots, s_T)$ with actions $a_{1:T-1}$, the induced EBM assigns

$$p_\theta(\tau) = \prod_{t=1}^{T-1} p_\theta(s_{t+1} \mid s_t, a_t) \propto \exp\left( -\sum_{t=1}^{T-1} E_\theta(s_t, a_t, s_{t+1}) \right).$$

A point prediction (MAP estimate) of the next state is obtained by minimizing the transition energy:

$$\hat{s}_{t+1} = \arg\min_{s'} E_\theta(s_t, a_t, s'). \tag{4}$$

This yields the most plausible single-step transition under the learned energy landscape.

**Training and Contrastive Loss**  Training an energy-based model by maximizing the log-likelihood,

$$\max_\theta \sum_i \log p_\theta(x^{(i)}),$$

is generally challenging because the partition function $Z(\theta)$ is intractable for most choices of energy functions. A common workaround is to model the *score function* of the distribution Hyvärinen (2005); Song & Ermon (2019) rather than the density itself. The score of a distribution $p_\theta(x)$ is

$$\underbrace{\nabla_x \log p_\theta(x)}_{\text{score}} = -\nabla_x E_\theta(x) + \underbrace{\nabla_x \log Z(\theta)}_{= 0 \text{ w.r.t. } x} = -\nabla_x E_\theta(x),$$

where the derivative of the normalization constant vanishes because it is independent of $x$. Thus, when computing gradients of the log-likelihood with respect to the data, only the negative energy term contributes. This implies that contrastive energy objectives (such as the loss introduced later in equation 5) can be optimized by gradient descent without ever evaluating or differentiating the partition function.

Because the partition function is intractable, EBMs are typically trained using contrastive learning methods that ensure observed (positive) transitions receive lower energy than artificially generated (negative) transitions. Let

$$x^+ = (s_t, a_t, s_{t+1})$$

denote a positive transition from the data. Negative transitions $x^-$ may be generated by Gaussian perturbations, predictions from a deterministic dynamics model, or short-run Langevin dynamics. A common contrastive loss is

$$\mathcal{L}_{\text{contrastive}} = E_\theta(x^+) + \lambda \log \sum_{x^-} \exp[-E_\theta(x^-)], \tag{5}$$

where $\lambda$ controls the balance between positive and negative samples. Negative transitions shape the energy landscape, encouraging the model to discriminate real transitions from dynamically implausible alternatives.

**Langevin sampling**  Because EBMs define unnormalized densities, sampling requires approximate methods. Langevin dynamics (Welling & Teh (2011), Du & Mordatch (2020)) performs stochastic gradient-based exploration of the energy landscape:

$$x^{(k+1)} = x^{(k)} - \frac{\epsilon}{2} \nabla_z E_\theta(x^{(k)}) + \sqrt{\epsilon} \, \eta_k, \qquad \eta_k \sim \mathcal{N}(0, I), \tag{6}$$

where $\epsilon$ is a step size and $\eta_k$ injects Gaussian noise at each step. Intuitively, Langevin dynamics can be understood as follows: if the energy function $E$ has a single minimum, starting from a random initial point $x^{(0)}$, the gradient term guides the sample toward the minimum. The stochastic term spreads the samples around the minimum, so repeated runs do not collapse to the exact same point. By performing multiple iterations and using many independent chains, one can generate diverse samples that approximately follow the underlying EBM density. From a theoretical perspective, Langevin diffusion and its discrete approximations are known to converge exponentially fast to the target distribution under suitable regularity conditions, providing a rigorous foundation for gradient-based sampling methods (Roberts & Tweedie, 1996).

In the context of transition energies, this procedure produces multiple plausible next states, which naturally provides a form of uncertainty quantification over predicted system behavior. Short-run Langevin sampling is typically used both to generate negative samples during training and to produce candidate transitions at inference time.

## 4 Model Architecture

We adopt a plug-in hybrid architecture inspired by the Time–Energy Model (TEM) framework introduced in Brusokas et al. (2025). A central feature of TEM is its modular design: a deterministic encoder–decoder forecaster provides an explicit transition model, while an energy-based model (EBM) is attached on top to evaluate the plausibility of candidate predictions, as shown in Fig. 1. This plug-in structure enables the energy model to interface with any standard forecasting backbone without modifying the underlying transition-learning mechanism. The model operates on state–action pairs $X := (s_t, a_t), Y := s_{t+1}$, where the task is both to predict the next state and to assess whether the proposed transition is consistent with the underlying system dynamics.

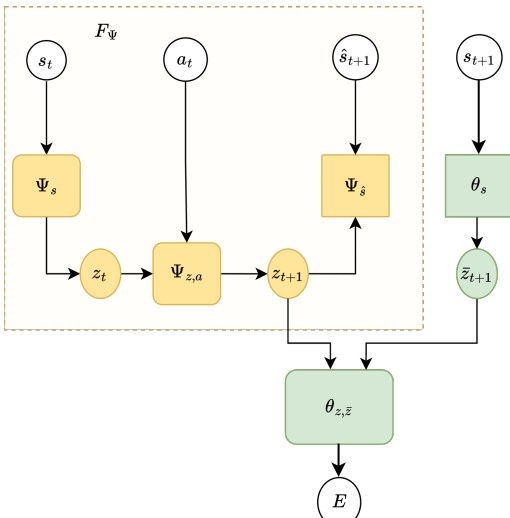

Figure 1: Hybrid architecture combining a deterministic forecaster with an energy-based model, adapted from Brusokas et al. (2025). The design enables the use of any state-of-the-art forecasting backbone with an accessible hidden state, while the deterministic model provides a reliable training signal that guides the EBM toward dynamically consistent transitions.

**Deterministic Transition Model** The deterministic component $F_\Psi$ serves as an explicit transition model operating in state–space form:
$$F_\Psi : \mathcal{X} \to \mathcal{Y}, \qquad \hat{s}_{t+1} = F_\Psi(s_t, a_t).$$
It is implemented as an encoder–decoder neural network,
$$\hat{Y} = \Psi_{\hat{s}}\big(\Psi_{z,a}(X)\big),$$

where $\Psi_{z,a}$ embeds the state–action pair into a latent representation and $\Psi_{\hat{s}}$ maps this latent state to the predicted next state.

During training, the deterministic model minimizes a mean-squared prediction error. Its primary roles are: (i) to provide a strong inductive bias for learning system dynamics, and (ii) to supply informative negative samples for the EBM during contrastive training.

**Energy-Based Transition Model**  The energy-based model $E_\theta$ evaluates the compatibility of a candidate transition $(s_t, a_t, s_{t+1})$ via

$$E_\theta : \mathcal{X} \times \mathcal{Y} \to \mathbb{R}.$$

Low energy values correspond to transitions that resemble observed system behavior, whereas high energies indicate implausible or dynamically inconsistent transitions LeCun et al. (2006).

Following Du et al. (2019), the energy function induces an unnormalized conditional distribution by using $-E_\theta(s_t, a_t, s_{t+1})$ as the energy term in equation 3. This yields a probabilistic interpretation of transitions without requiring explicit assumptions on noise structure or modality.

To couple the deterministic and energy-based components, the EBM shares the input encoder $\Psi_{z,a}$ with the forecaster while using its own output encoder $\theta_s$ and a joint decoder $\theta_{z\hat{z}}$:

$$E_\theta(X, Y) = \theta_{z\hat{z}}\big([\Psi_{z,a}(X), \theta_s(Y)]\big),$$

where $[\cdot, \cdot]$ denotes concatenation.

**Embedding Latent Variables and Avoiding Collapse**  A well-known challenge in energy-based encoder–decoder architectures is *representation collapse* in the latent space, where the encoders $z = \Psi_{z,a}(X)$ and $\hat{z} = \theta_s(Y)$ converge to constant or low-variance representations that trivially minimize the energy function.

In the proposed hybrid architecture, this issue is mitigated by coupling the EBM to a deterministic transition model that is straightforward to train. The supervised prediction loss imposed on $F_\Psi$ induces a meaningful latent structure in $\Psi_{z,a}(X)$, which in turn regularizes the EBM and discourages degenerate solutions.

In contrast, recent work by Balestriero and LeCun Balestriero & LeCun (2025) proposes an explicit regularization strategy to prevent collapse in purely self-supervised settings. Their LeJEPA framework enforces that latent representations follow an approximately isotropic Gaussian distribution, ensuring non-degenerate embeddings without relying on heuristics such as contrastive negatives or reconstruction losses.

While our approach relies on architectural coupling rather than explicit distributional regularization, both strategies address the same fundamental issue: maintaining expressive and informative latent spaces in predictive representation learning.

A central claim underlying the JEPA framework is that reconstructing raw observations (e.g., pixels in video data) is unnecessary, and that learning directly in latent space is sufficient for downstream tasks. In contrast, although our method performs self-supervised learning in a latent subspace, we explicitly retain full reconstruction of the observed data. For industrial process control applications, predicting the evolution of future sensor values is essential, as these quantities carry direct physical meaning and are required for monitoring, validation, and closed-loop control—properties that the latent space alone does not provide.

## 4.1 Training

The hybrid model (Fig. 1) is trained in two stages. In the first stage, the deterministic forecaster $F_\Psi$ is trained while keeping all EBM parameters fixed. In the second stage, the forecaster is frozen and the energy-based model $E_\theta$ is trained using a contrastive objective. The forecaster is optimized with a standard mean squared error (MSE) loss, and both training and validation errors decrease monotonically, as expected.

Monitoring the contrastive loss of the EBM alone proved insufficient to assess training progress, as its absolute value does not directly reflect improvements in the geometry of the learned energy landscape. To

obtain more informative diagnostics, we evaluate the energy of *perturbed transitions*:

$$E_\theta\left(s_{t+1}^{(n)}, s_t^{(m)} \;\middle|\; s_t^{(2:N)}, s_{t+1}^{(2:N)}, a_t\right),$$

which corresponds to substituting individual next states $s_{t+1}^{(n)}$ and current states $s_t^{(m)}$ while conditioning on the remaining batch elements and the action $a_t$.

Tracking the separation between the minimum-energy transition and these perturbed alternatives provides a more reliable indicator of training progress than the contrastive loss alone, as it directly reflects the model's ability to discriminate dynamically consistent transitions from implausible ones.

Figure 2 illustrates the evolution of the learned energy landscape for a single sample (sample 100) over epochs 1, 5, and 19. The red star indicates the energy minimum, which moves progressively closer to the true sample location as training proceeds. The lower-right panel shows the evolution of the energy gaps alongside the contrastive loss.

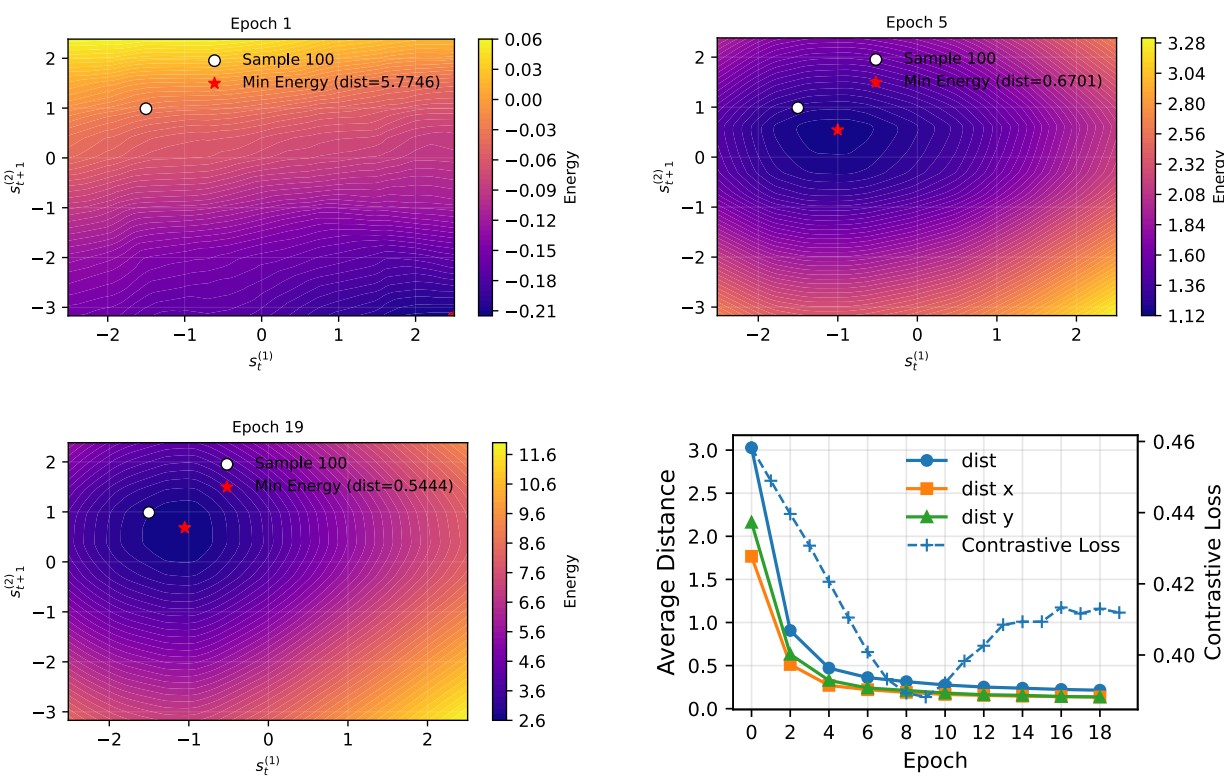

Figure 2: Evolution of the learned energy landscape during training for a single sample. Shown are transition energies at different epochs, together with diagnostics based on energy gaps and the contrastive loss. The lower-right panel shows the evolution of the energy gaps alongside the contrastive loss. The energy gaps are computed by evaluating the energy at perturbed state components along two coordinate directions, denoted as $x$-distance and $y$-distance. The minimum-energy prediction $\arg\min E_\theta$ is indicated by the red star in the first three panels and is compared to the corresponding ground-truth transition (white circle). This evaluation can be performed for all data points.

## 4.2 Prediction and Analysis

To estimate $\hat{s}_{t+1}$, we either use the deterministic transition model $F_\Psi$ or the minimum-energy (MAP) prediction of the EBM. In both cases, the deterministic prediction can be refined using short-run Langevin sampling conditioned on $(s_t, a_t)$ to obtain a probabilistic ensemble of plausible next states.

In our experiments, both initialization strategies produced similar results, which is expected because Langevin sampling is largely insensitive to the starting point provided it lies near a local minimum of the energy function $E_\theta$.

For analysis and visualization, predicted transitions are evaluated using the learned energy function in equation 2. This induces a *conditional energy landscape* under the learned dynamics:

$$\tilde{E}_\theta(s_t, a_t) = E_\theta\big(s_t, a_t, F_\Psi(s_t, a_t)\big). \tag{7}$$

where $\hat{s}_{t+1}$ denotes the deterministic prediction. High values of $\tilde{E}_\theta(s_t, a_t)$ indicate transitions that are inconsistent with the learned dynamics and therefore signal increased uncertainty or risk in multi-step forecasts. This landscape is visualized in Fig. 3 and Fig. 6.

### 4.2.1 Energy Landscape of a Second-Order System

Figure 3 compares learned energy landscapes and their associated gradient fields. Figure 3a corresponds to a conservative mass–spring system ($d = 0$), while Fig. 3b shows a damped system ($d = 0.05$). In the conservative case, the learned energy landscape captures the harmonic structure of the state variables—position $x_t$ and velocity $v_t$. Closed energy contours reflect the periodic nature of undamped oscillations. In contrast, the damped system produces spiraling trajectories that converge toward the origin, and the learned energy landscape correctly places its global minimum at $s_t = (0, 0)$. Measured trajectories are shown as blue dots, while recursive model rollouts obtained from the learned transition function $F_\Psi$ are shown as black crosses. This closed-loop simulation illustrates how the learned dynamics evolve within the energy landscape and its gradient field.

### 4.2.2 LTI system with ten states and two inputs.

We evaluate our method on a linear time-invariant (LTI) system with $n_x = 10$ states and two control inputs. A total of 800 datapoints were generated for training using randomly excited input signals. Measurement noise was added at 20 % of the signal standard deviation. Model quality is quantified using the normalized root mean square error (NRMSE) fit criterion defined in equation 8, which expresses how well the model reproduces the observed data relative to the mean of the signal. As a reference, we identify an LTI state-space model using the MATLAB function *ssest* with the correct model order $n_x = 10$. This baseline achieves an average fit of fit (%) = 80.01%, which can be viewed as an upper performance bound since the identification model matches the true data-generating structure. Our generic nonlinear EBM achieves an average fit (%) = 66.48% across all states. The corresponding simulation results are shown in Fig. 4, with a zoomed view of state 0 provided in Fig. 5. In these plots, black lines represent the measured trajectories, blue lines show the mean prediction obtained from Langevin sampling, and gray points correspond to the individual Langevin samples. For all experiments, we use a Langevin step size of `1e-2` and draw `100` samples, whose mean is used as the final prediction.

$$\text{fit } (\%) = 100 \left( 1 - \frac{\sqrt{\sum_{i=1}^{N}(y_i - \hat{y}_i)^2}}{\sqrt{\sum_{i=1}^{N}(y_i - \bar{y})^2}} \right) \tag{8}$$

where $y_i$ are the true values, $\hat{y}_i$ the predicted values, and $\bar{y}$ the mean of the true values.

**Energy Landscape and Stability Interpretation** We analyze the reduced energy function $\tilde{E}_\theta$ induced by the deterministic transition model and consider the associated vector field

$$\mathbf{F}(s_t) := -\nabla_{s_t} \tilde{E}_\theta(s_t, a_t),$$

where $\Delta$ denotes the Laplacian. The divergence of $\mathbf{F}$ characterizes local stability properties of the induced dynamics: negative divergence indicates locally contracting behavior, consistent with asymptotic stability near energy minima, while positive divergence corresponds to locally expanding and unstable regions. A

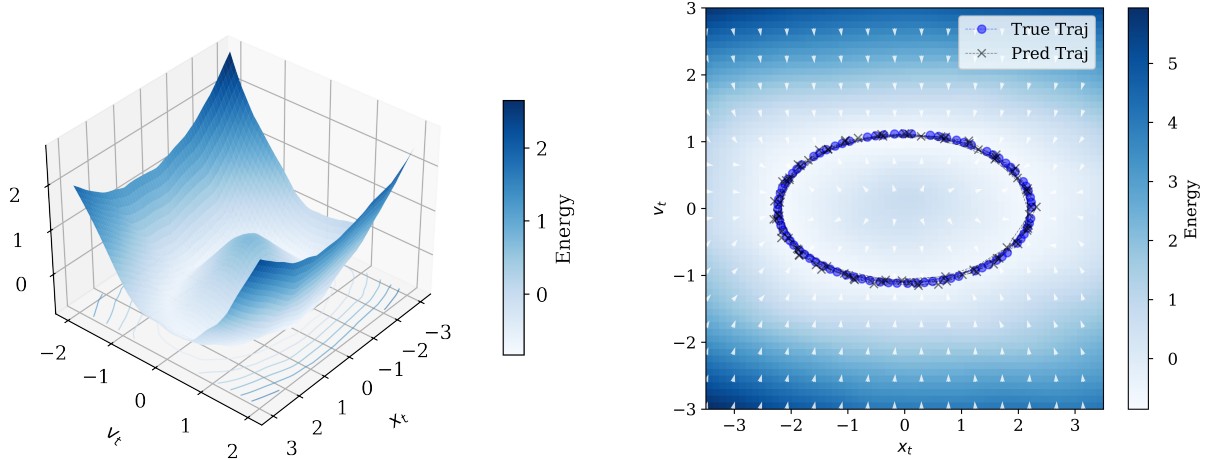

(a) Conservative system: learned energy landscape and corresponding gradient field.

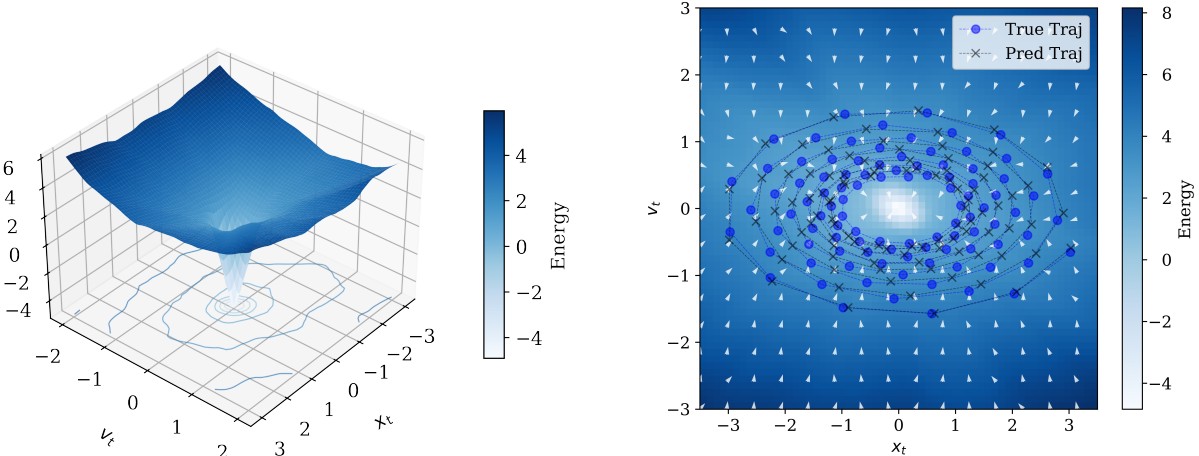

(b) Damped system: learned energy landscape and corresponding gradient field.

Figure 3: Comparison between conservative (a) and damped (b) systems.

divergence close to zero indicates approximately conservative, energy-preserving dynamics, as observed for undamped oscillatory systems.

**Setpoint Tracking via Energy Augmentation** As a brief remark, the proposed framework also admits a straightforward mechanism for setpoint regulation. To steer the system toward a desired setpoint $s_{\text{set}}$, we augment the transition energy with a quadratic tracking penalty:

$$E_{\text{track}}(s_t, a_t, s_{t+1}) = E_\theta(s_t, a_t, s_{t+1}) + \lambda \|s_{t+1} - s_{\text{set}}\|^2, \tag{9}$$

where $\lambda > 0$ controls the trade-off between transition plausibility and setpoint tracking.

The resulting implicit control law is

$$a_t^\star = \arg \min_{a_t} E_{\text{track}}\big(s_t, a_t, F_\Psi(s_t, a_t)\big), \tag{10}$$

which selects actions whose predicted transitions both remain in low-energy regions of the learned dynamics and move the system toward the desired operating point.

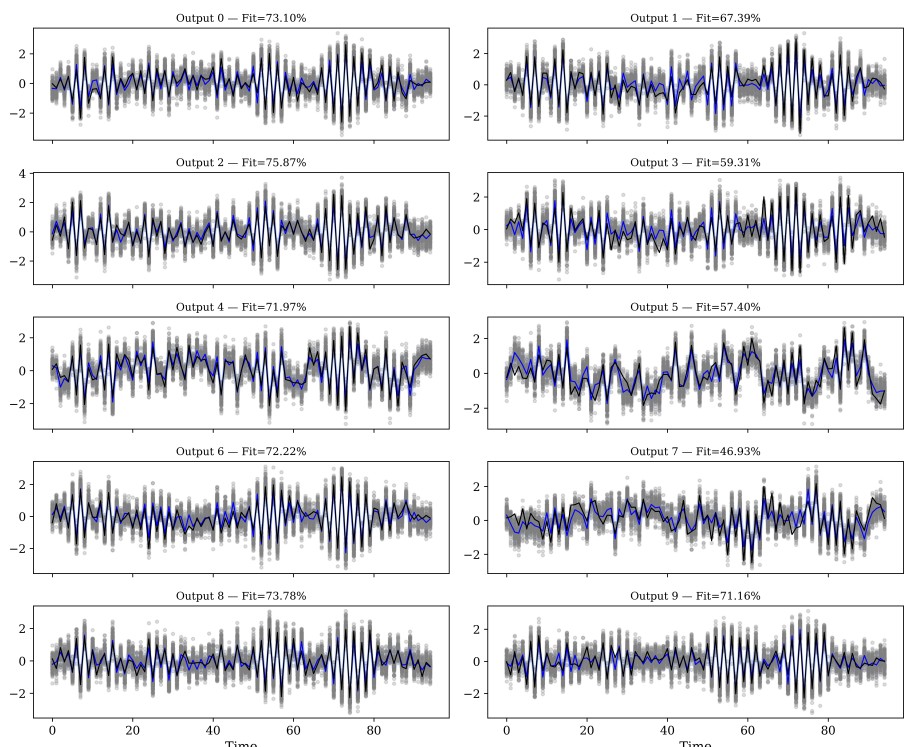

Figure 4: LTI validation. Simulation results for all 10 states. Black curves show the measured signals, blue curves show the mean prediction obtained from Langevin samples, and gray dots indicate the individual samples. The fit for each state is reported above the corresponding subplot, with an overall average of fit (%) = 66.48%. The Langevin sampling parameters are `step_size = 1e-2` and `n_samples = 100`. A zoomed view of state 0 is shown in Fig. 5.

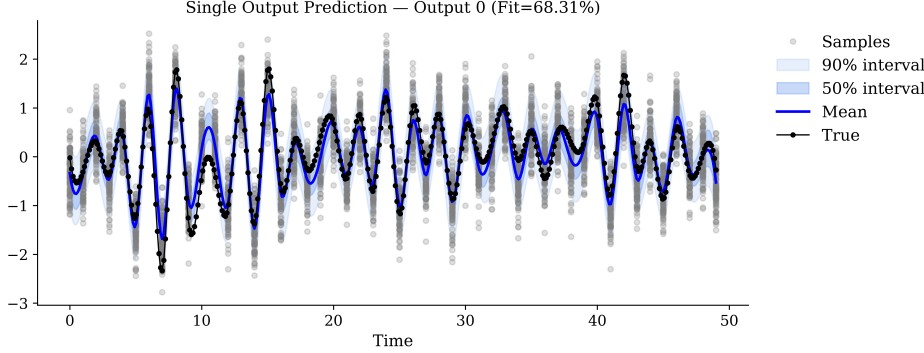

Figure 5: Zoomed view of output 0 from Fig. 4.

## 5   Conclusion and Outlook

In this work, we introduced the *Energy-Based Actionable World Model (EBAWM)*, a hybrid world-model architecture designed for recursive state evolution, long-horizon forecasting, and actionable decision-making in control settings. By combining a deterministic, action-conditioned state-transition model with an energy-based transition critic, EBAWM explicitly separates predictive structure from uncertainty evaluation. This design enables stable multi-step rollouts while preserving an interpretable and physically meaningful notion of state.

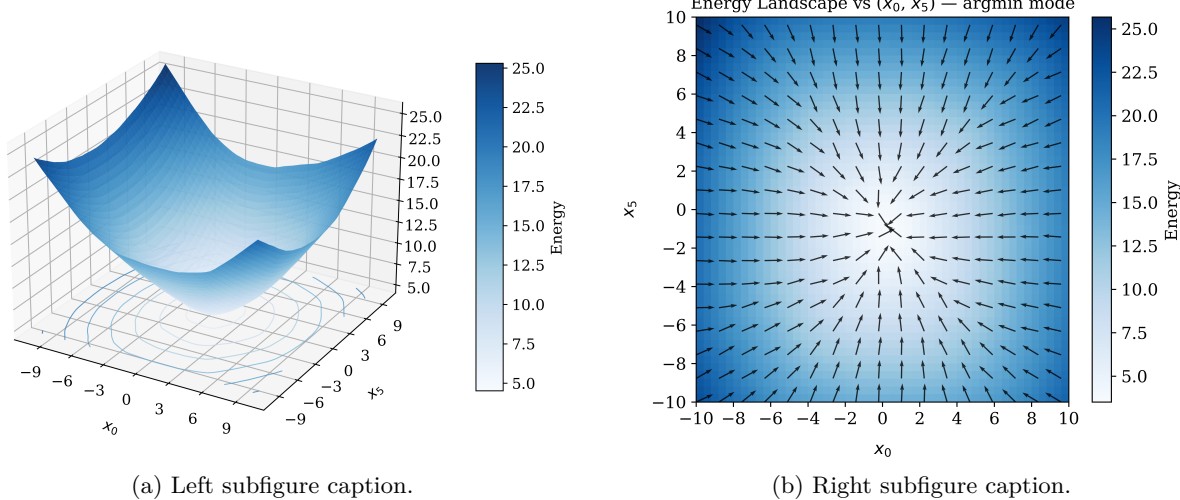

(a) Left subfigure caption.

(b) Right subfigure caption.

Figure 6: Energy field of the 10-dim LTI system plotted in two dimensions the other states have random values.

Unlike purely input–output predictors or implicit energy-only dynamics models, EBAWMM maintains an explicit state-space formulation that is compatible with RHC, model predictive control, and model-based reinforcement learning. The learned energy landscape provides an uncertainty-aware measure of transition plausibility, allowing the model to identify dynamically inconsistent or out-of-distribution behavior without relying on ensembles or parametric noise assumptions. Through experiments on nonlinear and linear dynamical systems, we demonstrated that the geometry of the energy landscape captures meaningful stability-related properties and supports both analysis and control via energy minimization.

In its current form, EBAWM propagates the *mean* of the predicted state forward in time, while uncertainty is evaluated locally through the energy function. This design choice is well aligned with classical state estimation and control: when transition uncertainty is approximately Gaussian, mean propagation is optimal in the minimum-variance sense and underpins widely used methods such as the Kalman filter and certainty-equivalent MPC. However, for strongly non-Gaussian, multi-modal, or asymmetric transition distributions, propagating only the mean may be insufficient to fully capture the evolution of uncertainty over long horizons. Addressing this limitation— for example by coupling energy-based evaluation with particle-based propagation, moment-matching beyond second order, or distributional rollout strategies—represents an important direction for future research.

Future work will focus on evaluating EBAWM on real-world industrial process data, where long time horizons, sensor noise, drift, bias, intermittent failures, and non-stationary operating conditions pose significant challenges. In such settings, the ability to maintain stable recursive state evolution while exposing actionable uncertainty signals is critical for safe and reliable deployment.

A particularly promising direction is *anomaly detection*. Because anomalous or faulty system behavior naturally corresponds to transitions that lie in high-energy regions of the learned landscape, EBAWM provides an intrinsic mechanism for detecting deviations from nominal dynamics. Integrating energy-based anomaly scores with control and monitoring pipelines could enable early fault detection, uncertainty-aware alarms, and graceful degradation strategies in safety-critical industrial systems.

Beyond anomaly detection, future extensions include incorporating explicit control constraints, exploring adaptive energy shaping for robust control, and studying formal stability guarantees induced by the learned energy function. Together, these directions position EBAWM as a step toward trustworthy, uncertainty-aware world models for industrial control and decision-making.

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
