# OpenReview forum: "Control-oriented Energy-Based Actionable World Model for Decision-Making and Process Control"
_TMLR — Rejected by TMLR_

### Review · Reviewer_zfjH · 2026-02-28

**Summary Of Contributions:**

The authors propose an energy-Based Actionable World Model (EBAWM) that leverages an explicit state-space formulation under an energy-based transition critic. The goal is to physically ground planning and simulation of long-horizon tasks via learning an energy-based critic that encodes states of low- and high-uncertainty. The proposed framework is supplemented with physically grounded priors, that prevent common mode collapse of energy-based systems. Finally, the authors intend leverage the learned energy landscape to implicit control and guide systems for closed-loop decision making.



**Strengths**

* The authors convincingly argue on the case of latent state-space models in order to preserve the long-term consistency of simulations and being able to continuously integrate novel information into latent space across timesteps. Their explicit energy guidance seems to be novel and well motivated.

* Motivation and underlying formalism are well presented and mostly self-contained. Experimental results are nicely displays and seem to demonstrate the working of the method.



**Weaknesses**

**1. Formalism and Related Work.** The authors spend varying portions of section 1-3 to define prior concepts and discuss related work. While the paper seems to cover relevant works and formalism overall, the different bits of information are scattered around the first 3 sections sections. E.g., for the discussion on EBMs, the introduction contains more references than the related work section, with the actual formalism only following in Sec. 3. Similarly, the authors note that their discussion on related work for EBM is not comprehensive. However, they rather broadly cite overviews ("Tutorial on EBM", "Introduction to Latent Variable Energy-Based Models") leaving out many specific works on EBM or energy-based optimization, and e.g. omitting references to foundational concepts such as MDPs. The authors might want to shift focus here and present a more detailed and contrasting discussion in the related work section.



**2. Novelty and Contributions.** Given rather extensive prior setup, the authors mention rather late that their approach adapts the existing Time–Energy Model (TEM) of Brusokas et al. (2025) and again continue with rather general discussions on EBM transitions and latent collapse without proper contrasting their improvements over the base model or discussing its structure and possible changes. As the prior model was only sparely discussed and my insight on this prior work is rather limited, I am still unsure which parts of the paper are present novel contributions and which do describe prior work. Similar to that, the presented text is quite 'wordy', e.g., mentioning "architectural coupling rather than explicit distributional regularization" and "we explicitly retain full reconstruction of the observed data" which is not supplemented by 'hard' equations/formalism that could help to better point-point exact differences.



**3. Experiments.** With the current state of experiments it is not possible to judge the actual contribution in contrast to similar methods and claims made by in the authors in the beginning. While the authors argue in the introduction for the insufficiency of RNNs/LSTMs or transformer based models, their experiments do, in no way, compare to any such methods. Similarly, given that their proposed EBAWM approach builds upon on a prior TEM method, I would have expected a comparison here, showcasing the relative improvement. Experiments are only carried out in comparison to a rather simple linear ssest MATLAB function on a linear time-dynamical system, which I would still expect to be solvable by a reasonably tuned JEPA model and similarly yields the questions of whether the repeatedly discussed representation collapse of prior models would actually occur under this setting. I'd like to mention that I do not per-se question the validity of the made claims above other architectures, but would like to see them supported via experiments.



**Minor: Overall Presentation.** The overall quality of figures could be improved if the plots where provided as vector graphics (e.g. as PDFs) rather than rasterized images, making text selectable. Similarly, most of the figure captions are rather short, making it hard to directly grasp their content. Figure 6 still contains "(a) Left subfigure caption." and "(b) Right subfigure caption." as subcaptions.

**Audience:**

No

**Audience Explanation:**

For the reasons mentions above I am having a hard time identifying the actual contributions of the paper and judging it's improvements over related work. While I believe that this might rather be a problem with the form of presentation and the conducted experiments than an inherent flaw of the approach, I can not imagine strong interest on the paper in its current form.

**Claims And Evidence:**

No

**Claims Explanation:**

The overall description of the model, as well as the lack of baseline comparisons makes it hard to judge the actual contribution of the paper. While the conducted experiments seem to generally align with made claims of a stable long-horizon simulation it is only evaluated under a rather simple/smoothly structured environment with no actual baselines compared.

**Requested Changes:**

As already discussed in the initial section I would like to recommend the authors to more concisely (and more formally) outline their contributions in contrast to prior work; particularly over the TEM approach that they seem to build upon.

While the authors provide an initial strong motivation on their approach over existing methods, the conducted experiments lack behind in providing experimental evidence for the made claims. Here I would like to recommend the authors to compare to comparable baselines, to substantiate the claimed benefits of their approach in terms of long-term stability and (the lack of) representation collapse.



**Minor**

*  Missing space in the abstract "learning.The deterministic"

---

### Review · Reviewer_Z6aj · 2026-03-03

**Summary Of Contributions:**

This paper proposes the Energy-Based Actionable World Model (EBAWM), a hybrid dynamics model that combines (i) a deterministic transition predictor $F_\Psi(s_t,a_t)$ trained with supervised prediction loss and (ii) an energy-based transition critic $E_\theta(s_t,a_t,s_{t+1})$ trained contrastively to assign low energy to observed transitions and higher energy to perturbed (negative) transitions.

For inference, the next state can be obtained either by the deterministic predictor or by minimizing the transition energy, optionally followed by short-run Langevin dynamics to generate a set of candidate next states. The paper also studies a reduced energy $\tilde E_\theta(s_t,a_t)=E_\theta(s_t,a_t,F_\Psi(s_t,a_t))$ and interprets its gradient field and curvature-based quantities as indicators of dynamical structure and stability-related behavior. Experiments are reported on a second-order mass–spring(-damper) system and a 10-dimensional LTI system.

**Strengths**
- Interesting use of an energy function as a transition plausibility / OOD signal, avoiding explicit parametric transition noise models and standard ensemble-based uncertainty heuristics.

**Weaknesses**
- Stability-related interpretations based on $\tilde E_\theta$ are presented more strongly than supported; the link to stability of the actual transition dynamics $s_{t+1}=F_\Psi(s_t,a_t)$ is not formally established and remains heuristic.
- The uncertainty signal is essentially single-step/local; the method does not propagate uncertainty through multi-step rollouts, despite long-horizon framing.
- Related work does not sufficiently situate the approach relative to modern uncertainty-aware model-based learning and long-horizon rollout methods.
- Empirical evaluation is limited (primarily linear systems) and lacks strong uncertainty-aware baselines (e.g., ensembles or probabilistic dynamics models).

**Audience:**

Yes

**Audience Explanation:**

The paper proposes an energy-based perspective on dynamics modeling and explores how the geometry of a learned energy landscape may encode transition plausibility and structural properties relevant to control. This conceptual angle is likely of interest to researchers working at the intersection of energy-based models, world models, and model-based reinforcement learning.

However, the empirical validation is limited to relatively simple systems, and the uncertainty-related claims are not benchmarked against modern uncertainty-aware MBRL methods. As such, while the underlying idea is potentially stimulating for parts of the TMLR audience, broader impact would require stronger empirical evaluation and clearer positioning within the current world-model literature.

**Broader Impact Concerns:**

There are no (direct) ethical implications of this work.

**Claims And Evidence:**

No

**Claims Explanation:**

**Over-general or unsupported claims**
Several central claims in the paper are stronger than what is theoretically established or empirically demonstrated.

**1. Claim: Energy Geometry and Stability**
The paper states that "the geometry of a learned energy landscape encodes meaningful dynamical structure and stability-related properties" and that analyzing gradients and curvature enables "stability analysis".

**Issue:**
The learned energy primarily reflects where state–action–next-state transitions are supported by data. Consequently, low-energy regions indicate transitions that are well represented in the dataset—not necessarily regions that are intrinsically stable in a dynamical systems sense.

If the data are collected under stabilizing control or concentrated around stable operating regimes, the learned energy landscape may indeed exhibit minima near stable equilibria. However, this would reflect properties of the data distribution rather than formal stability of the underlying dynamics $F_\Psi$. Conversely, if unstable regions are sufficiently sampled, the energy could also assign low values there, providing no meaningful stability signal.

Without establishing a formal relationship between the energy landscape and system dynamics (e.g., a Lyapunov-type decrease condition along trajectories of the learned dynamics), curvature and divergence of $\tilde E_\theta$ characterize properties of the energy geometry, not stability of the true or learned system.

As written, the stability interpretation appears stronger than warranted. It would be more precise to frame the energy landscape as encoding transition support under the training distribution, unless stronger theoretical guarantees are provided.

**2. Claim: Uncertainty-Aware Long-Horizon Forecasting.**
The method is positioned as a principled uncertainty-aware world model for long-horizon forecasting and control.

**Issue:**
Uncertainty is modeled locally at the single-step level via the energy function, while multi-step rollouts propagate only the mean prediction. No principled uncertainty propagation across horizons is performed. Consequently, characterizing the approach as uncertainty-aware for long-horizon forecasting overstates what is demonstrated.

In addition, the related work discussion in the MBRL context is limited (e.g., referring to MBPO, Janner et al., 2019), while recent advances specifically targeting principled uncertainty-aware long-horizon rollouts are not discussed, such as Frauenknecht et al., On Rollouts in Model-Based Reinforcement Learning (ICLR 2025), and Li et al., Uncertainty-Aware Robotic World Model Makes Offline Model-Based Reinforcement Learning Work on Real Robots (arXiv:2504.16680). The proposed method is not compared against any established uncertainty-aware long-horizon forecasting approaches, making it difficult to assess its relative strengths in the setting it claims to address.

**3. Claim: Validation as a General World Model**
The paper positions the proposed approach as a world model and states in the conclusion that it has been validated "through experiments on nonlinear and linear dynamical systems". This framing suggests empirical support across a range of dynamical regimes, potentially inclusing nonlinear systems relevant to model-based control and reinforcement learning.

**Issue:**
The experiments appear to be restricted to low-dimensional linear systems (e.g., an LTI system and a spring–mass–damper system in its standard linear formulation). As such, the empirical evaluation does not actually include genuinely nonlinear dynamics. Linear systems are comparatively simple to model and do not stress-test the method’s ability to capture nonlinear behaviors or complex dynamical structure. Consequently, the experimental evidence supports feasibility in linear settings but does not substantiate the broader “world model” framing or the claim of validation on nonlinear systems.

**Reproducibility Issues**
Several key implementation details are missing:
- Training of the deterministic model $F_\Psi$: single-step vs. multi-step loss, rollout horizon, optimizer and hyperparameters.
- Exact system definitions for the experiments: state-space equations, parameters, discretization method, and integration scheme.
- Given that the central claims depend on the learned energy landscape, and that this landscape is shaped by both data distribution and negative sampling strategy, these details are essential for reproducibility and interpretation:
    - How was the training data generated (input excitation, rollout length, reset conditions, noise model)?
    - How exactly are negative samples for contrastive learning constructed?

**Other Questions**
- The architecture employs latent encoders/decoders despite experiments being conducted on fully observable, low-dimensional Markovian systems. What is the motivation for introducing a latent representation in this setting, rather than modeling transitions directly in state space? An ablation comparing latent vs. direct state-space modeling would clarify whether the added complexity is necessary.

- The energy model uses a separate output encoder $\theta_s$ instead of sharing the state encoder $\Psi_s$ from the deterministic model. What is the rationale for not sharing parameters, given that both operate on the same state space?

**Requested Changes:**

**Critical for acceptance**

1. **Qualify or formally justify stability claims.**
   Clearly distinguish between stability of the true/learned dynamics $s_{t+1}=F_\Psi(s_t,a_t)$ and geometric properties of the induced energy landscape. Explicitly discuss the role of the data distribution (i.e., whether low-energy basins reflect intrinsic stability or merely data support). Substantially soften stability claims unless a stronger (theoretical) result is established connecting the energy landscape to true system stability.

2. **Align uncertainty claims with the method.**
   The model evaluates uncertainty locally but propagates only the mean state. Either provide explicit multi-step uncertainty propagation experiments or restrict claims to single-step uncertainty awareness rather than principled long-horizon uncertainty-aware forecasting.

3. **Align claims with empirical evidence.**
   Remove or revise statements claiming validation on nonlinear systems unless genuinely nonlinear experiments are added.

4. **Strengthen empirical validation.**
   Extend experiments beyond linear systems and include comparison against at least one uncertainty-aware baseline (e.g., ensemble or probabilistic dynamics model).

5. **Provide missing reproducability details.**
   Precisely specify: training procedure for $F_\Psi$ (single vs. multi-step), system equations and discretization, data generation, and negative sample construction (see reproducibility concerns above).

**Would strengthen the work**
1. Provide ablations on architectural design choices (latent vs. direct state modeling; shared vs. separate encoders $\theta_s$, $\Psi_s$).
2. Quantitatively analyze the relationship between energy and prediction error.
3. Demonstrate closed-loop control performance (e.g., setpoint tracking).

---

### Review · Reviewer_atzC · 2026-03-09

**Summary Of Contributions:**

The authors propose and Energy-Based Actionable World Model, meaning a model that for an input state $s_t$ and an action $a_t$, is able to predict $s_{t+1}$ using an energy-based model as probability description of the likelihood of the move.  The model does not operate directly in the space of states and actions, but rather in a latent space. Specifically, the EBM is fed with latent representations of a proposed transition, which is instead generated by a deterministic forecaster with an encoder–decoder structure. The authors describe each component of the model and then analyze some metrics during training. Finally, they test the model on a linear time-invariant system.

**Audience:**

No

**Audience Explanation:**

In its current form, the audience of the paper is, in my opinion, quite limited. Moreover, the validation of the model against benchmarks and/or a detailed theoretical or practical analysis is absent, making the contribution rather isolated. To provide a more compelling presentation, it would be necessary to compare the proposed method, at least theoretically, with comparable existing results. If not in the main text, this comparison should at least be included in the appendix. Moreover (see the points above and below), major revisions are still necessary at this stage.

**Broader Impact Concerns:**

World models are believed to be the next step of AI in some context, so a Broader Impact statement would be appreciated.

**Claims And Evidence:**

No

**Claims Explanation:**

The introduction of the paper and the claims are transparent, but both the presentation and the validation are not sufficient to support the objectives. In particular, the following list enumerates the main weaknesses:

 - Regarding the structure: there is a single long flow of explanations from the theory of EBMs to the conclusion. In this way, the reader is not facilitated in understanding what the novel contributions are, what belongs to the theoretical background, and what concerns the experiments. In my opinion, there should be a clear separation between theory and existing methods, and the novel proposals (model and experiments). Especially in a machine learning paper, such a separation is fundamental.
 - Regarding the introduction, the authors present three claims, but there are no hyperlinks to the sections in which they are discussed. If three results are claimed, it should be clear in which three sections they are treated.
 - Regarding the theory of EBMs, some comments that could be important—especially for a reader who is not an expert on the topic—are missing. First, there are works in the literature, for instance Carbone et al., NeurIPS 2023, but not only, where researchers manage to actually estimate the partition function during training. Secondly, Formula 5 is known to be affected by biases depending on the length of the chain used to generate negative samples. In my opinion, since EBMs are the core of the model, a more detailed discussion of their limitations should be presented, as well as an explanation of why these issues do not arise in this case. In other words, the presentation of EBMs does not address the well-known limitations of these generative models.
 - Regarding the model: the authors mention that the model is adapted from “Brusokas, 2025”, but they do not discuss to what extent their proposal differs from that work, not even in the appendix. It is difficult to evaluate the novelty of the work if the only reference to the prior method is the word “adapted”.
 - Section 4.2.1 is somewhat confusing. The only occurrences of the word “damped” are related to that section, but there is no previous discussion of the dynamics that need to be analyzed. For example, what does d represent?
 - Regarding the experiments: first of all, for a machine learning paper it is, in my opinion, necessary to release the code during the review phase, especially if the main claims are related to a newly proposed model. Secondly, the authors do not explain what an LTI system is. I believe that omitting a description of the model may be acceptable for extremely well-known benchmarks such as CIFAR-10 or MNIST, but in this relatively novel area of world models it is important to help the reader understand the experiment being performed. For instance, are there other benchmark results or evaluation metrics? Alternatively, if the objective is just a proof of concept, a detailed analysis of failure modes should be presented. If only one experiment is provided, without references or explanations in the appendix, an experimental machine learning paper becomes quite weak in terms of convincing and clear evidence.

In general, I found that the positioning of the paper is unclear. If the objective is to propose a model that competes with existing methods, then clear benchmarking should be provided. If it is a proof of concept, then a very detailed analysis (even if limited to the LTI case) should be presented, both theoretically and experimentally. At the moment, neither of these two directions is clearly pursued.

**Requested Changes:**

- The sentence *“where ∆ denotes the Laplacian”* at the end of page 9 appears after an equation without an equation number and without a Laplacian.

- A strong restructuring is necessary: please separate the theory and existing models from the novel proposal and the experiments, and improve the section organization, possibly into two or three major sections. Also try to associate the claims in the introduction with precise sections.

- Complete the theory section on EBMs by adding more recent references, especially regarding the partition function and the problems that arise in EBM training in relation to contrastive learning, and explain why these issues are not relevant in the context of this paper.

- Since you state that the model is adapted from *“Brusokas, 2025”*, please present that model in detail (at least in the appendix) and highlight your novel contributions in comparison to it. This is fundamental for evaluating the originality of the work.

- Clarify what the experiments in Section 4.2.1 are about, and possibly support them with a theoretical analysis.

- Please provide an explanation of LTI systems. If the objective is benchmarking, include results from existing methods for comparison. If not, provide a more detailed analysis of the behavior of your model, including failure modes and particular situations that may be of interest, or include additional experiments to validate your proposal.

---

### Decision · Action_Editor_YYt9 · 2026-04-16

**Recommendation:** Reject

**Additional Comments:**

The manuscript proposes a model for predicting states of a dynamical process; the model is termed "Energy-Based Actionable World Model (EBAWM)". The model combines a deterministic transition predictor, trained with supervised prediction loss, with an energy-based transition critic, trained contrastively to assign low energy to observed transitions and higher energy to perturbed (negative) transitions. The energy function thus serves to capture uncertainty, where high energy shall indicate inconsistent or out-of-distribution behavior. The model is intended for long-horizon prediction and use in decision making. Numerical results are reported on a second-order mass–spring(-damper) system and a 10-dimensional linear time-invariant system.

---

Three reviews have been collected. While the reviewers find the proposal and use of energy functions interesting, they all point out significant issues and weaknesses in the manuscript and reported results. The reviewers uniformly agree that the manuscript cannot be accepted.

---

**Review process:** The reviewers provided valuable comments. The authors did not submit a rebuttal or revised version of the manuscript. Hence, all concerns and requests for changes by the reviewers remained unaddressed.

**Summary of evaluation:** Based on the reviews and raised issues, the manuscript is not suitable for publication in TMLR.

**Audience:**

No

**Audience Explanation:**

All reviewers have doubts whether the manuscript in its current state is of interest. Key aspects that limit interest by a potential audience pertain to very limited validation, overclaims, and of lack of theoretical analysis and benchmarking/comparison with existing models. Furthermore, the authors did not respond to any of the concerns or requests.

**Claims And Evidence:**

No

**Claims Explanation:**

All three reviewers agree that the claims are not supported by sufficient evidence. In particular, the reviewers point out serious concerns regarding the manuscript's structure and presentation, positioning with respect to the literature (in general and in technical details), insufficient empirical results and comparisons, and reproducibility issues. Furthermore, central claims are not backed up by proper analysis and results (e.g., link between energy geometry and stability; no proper uncertainty propagation while claiming uncertainty-aware long-horizon forecasting; insufficient validation for "world model" claim).